# Deep Learning Approach for Detection of Underground Natural Gas Micro-Leakage Using Infrared Thermal Images

**DOI:** 10.3390/s22145322

**Published:** 2022-07-16

**Authors:** Kangni Xiong, Jinbao Jiang, Yingyang Pan, Yande Yang, Xuhui Chen, Zijian Yu

**Affiliations:** 1College of Geoscience and Surveying Engineering, China University of Mining and Technology, Beijing 100083, China; bqt1800205063@student.cumtb.edu.cn (K.X.); bqt1700205018@student.cumtb.edu.cn (Y.P.); zqt17002050101T@student.cumtb.edu.cn (Y.Y.); zqt1900205112g@student.cumtb.edu.cn (Z.Y.); 2Satellite Application Center for Ecology and Environment, Beijing 100094, China; tbp150202039@student.cumtb.edu.cn

**Keywords:** natural gas leakage, vegetation stress, infrared thermal image, convolution neural network, transfer learning

## Abstract

The leakage of underground natural gas has a negative impact on the environment and safety. Trace amounts of gas leak concentration cannot reach the threshold for direct detection. The low concentration of natural gas can cause changes in surface vegetation, so remote sensing can be used to detect micro-leakage indirectly. This study used infrared thermal imaging combined with deep learning methods to detect natural gas micro-leakage areas and revealed the different canopy temperature characteristics of four vegetation varieties (grass, soybean, corn and wheat) under natural gas stress from 2017 to 2019. The correlation analysis between natural gas concentration and canopy temperature showed that the canopy temperature of vegetation increased under gas stress. A GoogLeNet model with Bilinear pooling (GLNB) was proposed for the classification of different vegetation varieties under natural gas micro-leakage stress. Further, transfer learning is used to improve the model training process and classification efficiency. The proposed methods achieved 95.33% average accuracy, 95.02% average recall and 95.52% average specificity of stress classification for four vegetation varieties. Finally, based on Grad-Cam and the quasi-circular spatial distribution rules of gas stressed areas, the range of natural gas micro-leakage stress areas under different vegetation and stress durations was detected. Taken together, this study demonstrated the potential of using thermal infrared imaging and deep learning in identifying gas-stressed vegetation, which was of great value for detecting the location of natural gas micro-leakage.

## 1. Introduction

Natural gas, as a clean energy source, has continued to attain sustained growth in consumption in recent years [1,2]. With the popularity of natural gas, the increased risk of natural gas leakage during transport and distribution has also attracted widespread attention. As the main mode of natural gas transportation, the buried pipeline may be destroyed to cause leakage by corrosion, construction damage, anthropic activities, and natural disasters [3,4], which subsequently leads to personal injury, financial losses, security, and ecological issues [5]. Therefore, it is of great significance to detect natural gas leakage timely and accurately.

Traditional natural gas leakage detection mainly includes manual inspection [6], sound monitoring [7], gas sampling, soil detection [8], flow monitoring [9], software-based dynamic modeling monitoring [10], etc. However, these methods not only consume a lot of manpower, financial resources and time, but also cause certain damage to pipelines, gas storage, and the surrounding environment [11]. Infrared thermography is increasingly used in non-destructive condition monitoring services as a less labor-intensive and non-contact technique to assess vegetation stress by leaf canopy temperature [12,13,14,15]. Canopy temperature has been widely used to detect stress because the closing of the stomas on the leaves is controlled to retain water during the stress, resulting in changes in stomatal conductance, transpiration, and leaf temperature [16,17,18,19,20,21]. Some research found that the canopy temperature between the stressed vegetation and healthy vegetation was different [22,23,24,25]. Joalland et al. [23] found that sugar beets infected by nematodes exhibited the leaf temperature higher than healthy plants. Tian et al. [26] found that the canopy temperature of vegetation under salt stress increased significantly with increasing salinity in the soil. Maes et al. [24] found that the canopy temperature of mistletoe-infected trees was significantly higher than that of uninfected trees. Therefore, canopy temperature can be supported as a viable indicator of vegetation health [27,28]. Currently, infrared thermography is widely used in water stress [27,29,30,31,32,33,34], salinity stress [26], metal stress [35] and crop disease [36,37]. But little attention is focused on natural gas stress.

The study of canopy temperature is mostly based on the average data measured from areas or points, which may be more useful than individual leaf temperatures in detecting vegetation stress [38]. However, it must also be considered that the canopy temperature of stressed vegetation exhibits great spatial variability [39]. The development of image processing and deep learning technology provides a new method for extracting canopy temperature and predicting vegetation stress [40]. Deep learning methods represented by convolutional neural networks (CNN) avoid the labor-intensive feature extraction process, which show great superiority. Meanwhile, deep learning methods can extract more accurate spatial thermal features of the vegetation canopy from infrared thermal images to achieve higher classification ability, which has been widely used in precision agriculture [39,40]. Maimaitijiang et al. [41] fused multi-source remote sensing data including infrared thermal images to predict soybean yield by the deep learning methods. King and Shellie [42] proposed neural network models to detect the water stress of grapes by the leaf temperature. Deep learning methods provide indicators for vegetation growth monitoring under different environmental conditions. In addition, transfer learning as a method of deep learning can effectively use the features learned from large data sets for applications with small training data sets, which can solve the problem that deep learning models need enough training data [43,44,45]. Jiang et al. [46] proposed a method based on deep learning and thermal imaging, which transfer learning was used to pretrain mask RCNN network for plant segmentation and temperature extraction. Therefore, it is valuable to extract vegetation canopy information from infrared thermal images using a deep learning approach for vegetation stress detection.

The aim of this study was to detect natural gas micro-leakage stress areas in four vegetation varieties using infrared thermal images with deep learning methods. The objectives were as follows: (a) to explore the correlation between gas concentration and vegetation canopy temperature under natural gas micro-leakage stress; (b) to classify the natural gas stress of four species of vegetation based on the CNN; (c) to improve the efficiency of the classification model based on transfer learning; (d) to detect natural gas micro-leakage areas and time series analysis.

## 2. Materials and Methods

### 2.1. Study Area and Site Description

The impact of natural gas micro-leakage on vegetation was studied by field simulation experiments to indirectly detect natural gas micro-leakage areas. The experiment field was located in Daxing District, Beijing, with coordinates of 39°39′2.56″ N, 116°34′33.10″ E. As shown in Figure 1, the data were obtained from two different types of plots. The first measurement area was a 20 m × 5 m plot with eight gas leakage points and planted grass. The second measurement area was set up with twenty-four plots of 2.5 m × 2.5 m. Wheat, soybean, grass, and corn were planted alternately with seasons. There were eight plots for each plant species, including four control plots and four experimental plots. Each plot was separated by a narrow road with a width of 50 cm. A 1.5 m wide road was reserved between each row of plots as the data acquisition channel. The plots with odd numbers represented the experimental plots while even numbers represented control plots without natural gas. The leakage rate of each experimental plot was controlled in 1 L/min by the gas mass flow controller for the whole day.

This experiment lasted from 2017 to 2019. Each plot received the same standard agronomic practices such as watering, fertilization, weeding, pest control and uniform seedling during the experiment, which can ensure that the vegetation was unacted on the other controllable factors.

### 2.2. Test Setup and Experimental Procedure

In this experiment, ten gas concentration sampling points were evenly distributed in the experimental plot to determine the concentration of natural gas in the soil. Three hollow PVC pipes were inserted into each sampling point, with a depth of 60 cm underground. Since the main component of natural gas is methane, a GXH-3050E gas analyzer (JunFangLiHua, Beijing, China) was used to collect the methane concentration of each point every three days during the whole experimental period (Figure 2a). The average of the methane concentration at each point was taken to draw the contour map of methane volume fraction by MATLAB (Mathworks, Natick, MA, USA), which was used to clarify the spatial distribution law of natural gas micro-leakage.

Fluke ti55ft thermal imager (Fluke Corporation, Washington, DC, USA) was used for data acquisition (Figure 2b). The thermal imager can obtain both infrared thermal and visible images with a 320 × 240 fixed pixel array. The thermal sensitivity is less than or equal to 0.05 °C, and the temperature range is from −20 °C to 600 °C. Images were taken from 9:00–11:00 AM local time in cloudless weather and steady solar radiation intensity. The lens of the thermal imager should be kept vertically downward. The vertical distance between the lens and the vegetation canopy was 5 m. Each plot was photographed for approximately 30 s. Two or three images of each plot were collected to obtain the infrared thermal images of the control or experimental plot.

### 2.3. Image Pre-Processing

The infrared thermal images taken in the field were processed in smartview4.3 software (Fluke Corporation, Washington, DC, USA). The data collected from field experiments needed to be removed for singular values before comparative analysis due to the influence of metal marker points. In addition, deep learning approaches require a large number of training samples to learn the weights and parameters in the training phase [47,48]. Due to the limited infrared thermal data collected in the field, the training dataset was enhanced by using random flip, random angle rotation, cropping, color transformation, and noise transformation. Finally, 7571 images were obtained for model training. 1820 images and 428 images were used as the validation and test sets, respectively.

### 2.4. GLNB Architectures for Classification

Compared with the traditional model, the deep learning approaches provide better results for the recognition and classification of agricultural images [49]. In this paper, the GLNB model incorporating GoogLeNet and Bilinear pooling was proposed to detect stressed vegetation. The detailed structure is shown in Figure 3. GoogLeNet structure consisted of nine Inception modules connected in series. The structure of Inception is shown in Figure 4. The Inception module added 1 × 1 convolution before 3 × 3 convolution and 5 × 5 convolution, and 1 × 1 convolutional layer after 3 × 3 max pooling. This structure made more efficient use of computational resources by incorporating feature information of different scales, in which different features were extracted with the same amount of computation to improve the training results. Furthermore, the bilinear pool captured the relationship between features by obtaining the outer product of two vectors to model the higher-order statistical information of the features [50]. The outer product captured the pairwise correlation between feature channels in a translation-invariant manner. Bilinear pooling could be optimized end-to-end, which provided a stronger feature representation than linear models. The GLNB model was trained with infrared thermal images for 200 epochs in Python 3.7 using PyTorch backend.

### 2.5. Implementation of the Transfer Learning

Transfer learning pre-trains the model and retains the parameters of the model. Then, a small amount of target data is used to retrain the fully connected layer. This approach can optimize the learning efficiency of the model and improves the model performance [51,52]. However, not all data are equally informative. In this article, data containing similar partial information were selected as samples for transfer learning. The visible images in this experiment, ImageNet [53] and the public infrared thermal dataset [54,55,56,57] were used as the transfer learning training set for pre-training the GLNB model. Then, the parameters were saved and imported to the model with fine-tuning.

### 2.6. Visual Explanations and Gradient-Based Localization

Gradient-weighted Class Activation Mapping (Grad-CAM) assigns the gradient information of the last convolution layer to each neuron and generates a location map to highlight the important region in the image, which is the basis of output prediction [58]. The last convolution layer should capture the picture features more specifically. Grad-CAM uses the gradient information from the last convolution layer in CNN to understand the importance of each neuron for category recognition. For category *c*, the weight ($\alpha_{k}^{c}$) of the kth neuron in the last convolutional layer is calculated as follows:(1)$$\alpha_{k}^{c}=\frac{1}{Z}\underset{i}{\sum }\underset{j}{\sum }\frac{\partial y^{c}}{\partial A_{ij}^{k}}$$

Equation (1) can be intuitively understood as: take the input ($y^{c}$) of the softmax layer and calculate the partial derivative of the ijth input pixel value of the kth neuron in the last convolution layer. Then, the kth neuron weight ($\alpha_{k}^{c}$) in the last convolution layer is obtained by the global average pooling. For category *c*, the classification localization map ($L_{Grad-CAM}^{c}$) is calculated as follows:(2)$$L_{Grad-CAM}^{c}=ReLU\;\left(\underset{k}{\sum }\alpha_{k}^{c}A^{k}\right)$$

Equation (2) can be intuitively understood as: the input of different neurons in the last convolutional layer are multiplied by their respective neuron weights. Then the corresponding values of all neurons are summed. The ReLU retains the pixel values that have a positive effect on the classification and suppresses the pixel values that have a negative effect on the classification, and then a heat map, i.e., the classification localization map ($L_{Grad-CAM}^{c}$) is obtained.

### 2.7. Performance Evaluation

In this paper, the confusion matrix was used to evaluate the classification performance of the CNN model. Accuracy represents the proportion of all correctly matched samples to all samples. However, when the number among different samples is unbalanced, there is a great error in using the accuracy as the index. Multiple evaluation indexes being used to comprehensively evaluate the model can more truly reflect the effect of the model. Therefore, the accuracy, recall and specificity were used to evaluate the performance of the model in this study. The following is the formula:(3)$$Accuracy=\frac{TP+TN}{TP+TN+FP+FN}$$
(4)$$Recall=\frac{TP}{TP+FN}\;$$
(5)$$Specificity=\frac{TN}{TN+FP}\;$$
where, True Positive (TP) represents correctly identified experiment plots; False Negative (FN) indicates unsuccessfully identified experiment plots; False Positive (FP) represents control plots identified as experiment plots; and True Negative (TN) indicates correctly identified control plots.

## 3. Results and Discussion

### 3.1. Stress Symptoms

The infrared thermal image could not cover the whole plot because of the small field angle of the infrared thermal imager. But it could cover the main stress area in the experimental plot. Infrared thermal images of the four vegetation in the experimental plot and the control plot are shown in Figure 5, which are grass (*Cynodon dactylon* L.), soybean (*Glycine max* Merr.), corn (*Zea mays* L.) and wheat (*Triticum aestivum* L.), respectively. Soybean, wheat and corn are one of the main crops in China. Grass is a good plant for Embankment Consolidation and soil conservation, which is often used to pave lawn for urban greening. The four types of vegetation have different characteristics and are widely planted in China.

As shown in Figure 5, two cross-sectional lines NE-SW and NW-SE were selected on the image. The temperature value of cross-sectional pixels was extracted to draw the temperature curve for analysis. Figure 6 clearly shows that the temperature variation of NE-SW and NW-SE in the experimental plot was high in the middle and low on both sides, while the temperature in the control plot showed a stable trend. In addition, the mean, standard deviation, maximum and minimum values of temperature in the profiles were counted to compare the differences of vegetation temperature between the experiment plots and control plots.

According to the statistical characteristic values of temperature distribution (Table 1), the mean, standard deviation and maximum temperature of the experimental plot were higher than those of the control plot, but the minimum temperature was close to equal. This indicated that the temperature fluctuation in the experimental plot was more drastic, and little temperature fluctuation in the control plot. The maximum temperature difference of the experimental plots among the four types of vegetation was greater than 10 °C, and the average temperature difference between the experimental plot and the control plot ranged from 3.10 °C to 3.55 °C. Combined with Figure 6, different vegetation affected by natural gas micro-leakage stress produced similar temperature characteristics and spatial characteristics, i.e., the vegetation temperature in the central area affected by natural gas leakage was higher than that in the edge, and the temperature values showed spatial characteristics of high in the middle and low in the sides. This is similar to the research of Noomen [59]. This indicated that the poor growth of vegetation under the stress of natural gas leakage led to great changes in its temperature field, which was consistent with the vegetation growth observed in the field.

The infrared thermal image was matched and superimposed with the gas concentration distribution map as shown in Figure 7a. It was found that the concentration distribution contours of the natural gas diffusing in the soil was roughly quasi-circular. This was because the leakage of natural gas led to the loss of oxygen in the soil, which inhibited the function of roots [60]. Due to the long-term impact of high concentration of natural gas, the vegetation respiration in the area where the gas concentration was more than 30% was weakened and showed stress symptoms such as Leaf Chlorosis and growth retardation, which made the canopy temperature of stressed vegetation higher than that of healthy vegetation and showed a quasi-circular stress area in space. The 236 random points in the superposition part of the infrared thermal image and gas concentration distribution map were selected to extract the corresponding temperature values and gas concentration values for correlation analysis. As shown in Figure 7b, the temperature of the experiment plot was positively correlated with the gas concentration and the root mean squared error was 0.87. Therefore, the canopy temperature distribution characteristics of vegetation were consistent with the gas concentration distribution, i.e., both showed quasi-circular distribution. This further confirmed that the difference in canopy temperature was caused by the natural gas leakage. In addition, the range of stressed vegetation areas might vary due to different plant tolerances to natural gas, but the spatial features remained similar. Therefore, this kind of circular spatial feature can be used for the identification of vegetation stressed by natural gas micro-leakage.

### 3.2. Performance of the GLNB Method

Infrared thermal images of four plants were input into the GLNB model for 200 epochs training. It was found by many attempts that a large learning rate could cause the loss curve to oscillate and the model failed to fit, while a small learning rate could cause slow convergence of the loss curve and overfitting of the model. Therefore, 0.0001 was applied to the proposed model as the appropriate learning rate. Adam made the model converge faster by comparing the performance between Stochastic Gradient Descent (SGD) and Adam, so Adam was applied to the proposed model. In addition, four classical CNN models (AlexNet [61], VGG16 [62], GoogLeNet [63] and ResNet [64]) were developed for comparison. Results as shown in Table 2, the five CNN models effectively identified stressed vegetation plots under natural gas micro-leakage. The proposed GLNB model with 91.59% average classification accuracy achieved better results than other models. Furthermore, the recall and the specificity of average per class in the proposed model were 92.32% and 91.07%, respectively, which were higher than other models (AlexNet: 90.16% and 86.09%, VGG16: 87.03% and 88.40%, GoogLeNet: 85.87% and 86.97%, and ResNet: 86.51% and 83.59%). This indicated that the proposed GLNB model performed superior to identify the stressed vegetation plots under the natural gas micro-leakage. This is because the model includes Inception structure and bilinear pooling, which integrates feature information of different scales and makes use of the difference of second-order statistical information to improve the performance of the model.

In the proposed GLNB model, grass got a better classification accuracy (95.33%) than other kinds of plants (soybean: 90.65%, corn: 88.79%, and wheat: 91.59%). Similar results were obtained in the other four classical models. Grass had a higher stress recognition rate rather than soybean, corn and wheat, which showed more sensitive to the nature gas. There may be the following reasons. The different canopy structure makes the irregular stress shape of corn, soybean and wheat, which makes large identification errors. Corns are taller with sparse planting spacing, which is easy to produce shadow and causes recognition error. Wheats have tufted and thin stalks, which is easy to cause uneven canopy coverage. Soybeans have strong rhizomes with large coverage of single plant. Meanwhile, the soybean rhizobia are easy to affect the stress symptoms of natural gas. Ferchichi [65] found that soybean rhizobia promoted vegetation growth and helped plants resist the impact of adverse environment, which made stress symptoms less pronounced in soybean. Grass canopy is denser and more uniformly covered, while the shallow root system is more sensitive to the increased gas concentration in the soil. In contrast, other plants have the more developed root system than grass, which could better adapt to the environment. Other researchers had done similar experiments on vegetation stress. Smith, K.L [66] found that wheat and beans were not affected when gas was delivered to well-developed wheat and beans. This might be because the plant had developed a complete root system that allowed its roots to obtain nutrients and water from outside the gas-affected area. Therefore, the grass can be used as a good indicator to detect natural gas micro-leakage areas.

### 3.3. Comparison with the Transfer Learning

The GLNB-based transfer learning model (GLNB-TL) was developed to use public datasets as the training set to pre-train the model and obtained the corresponding model structure and weights. The iterative curves of the GLNB and GLNB-TL during training are shown in Figure 8. The accuracy and loss curves were used to determine the fitting situation of the training model to optimize the model parameters. As the number of iterations increased, the accuracy of the training and validation sets gradually rose while the loss value gradually dropped. Finally, both accuracy and loss curves tended to be stable. Comparing with the iterative curves of the GLNB and GLNB-TL, it was found that GLNB-TL could converge with fewer epochs and obtained lower loss value and higher accuracy. Therefore, transfer learning makes the training model converge quickly and improved the training efficiency.

The test results of GLNB and GLNB-TL are presented in Table 3. The GLNB-TL achieved the average classification accuracy of 95.33%, which was higher than the GLNB model. The recall and specificity of average per class also performed well with the 95.02% and 95.52% classification scores, respectively. Grass had a better classification accuracy (98.13%) than other kinds of plants (soybean: 96.26%, corn: 91.59%, and wheat: 95.33%). Comparing the performance between GLNB and GLNB-TL, it was found that the GLNB-TL model had the better classification performance in a variety of plants, which showed that the transfer learning method improved the performance and generalization ability of the model. This is because the weights of the pre-trained model store key information to identify the basic feature of the stressed vegetation in the proposed model. The transfer learning could customize its own weights for identifying the stressed vegetation in the infrared thermal dataset faster and easier. Therefore, transfer learning has application value in the identification of the natural gas micro-leakage stressed vegetation.

### 3.4. Detection and Time-Series Analysis

Grad-CAM was used to generate the heatmap of class activation in this paper, which visualized the classification features of the input image. The last convolutional layer captured more specific features of the image, so Grad-CAM used the gradient information from the final convolutional layer of the model to generate heat map to understand the importance of each neuron for category recognition. Figure 9 shows the visualization results. The proposed GLNB model in this paper correctly extracted the features of stressed vegetation from the infrared thermal image and located the micro-leakage area of natural gas. Therefore, this proposed model can identify stressed vegetation and detect natural gas micro-leakage.

Infrared thermal images of five periods were selected for each vegetation type. The stressed vegetation areas caused by natural gas micro-leakage were extracted by Grad-CAM and segmented based on the adaptive threshold method. Vegetation stress areas were consistent with natural gas micro-leakage areas, which had been demonstrated in Section 3.1. So the natural gas micro-leakage areas were located by drawing fitted circles based on the quasi-circular spatial distribution rules of gas stressed areas. The temporal and spatial variation of the range of stressed vegetation areas under natural gas micro-leakage is shown in Figure 10. Due to the influence of continuous natural gas micro-leakage, grass and wheat showed growth retardation and even death. The concentration of natural gas in soil gradually accumulated and diffused with time, so the stress range of grass and wheat showed a trend of gradually expanding. The range of stress in soybean showed a gradual decrease with the duration of stress. This is because both the canopy and the root system of the soybean seedling stage were not well developed, which leaded to its sensitive to the external environment. Therefore, the stress range of soybean was large in the early stage. With the growth and development of soybean, the developed root systems leaded to the stable growth situation, which improved the tolerance to the adverse external growth environment. Meanwhile, the effect of soybean rhizobia was also one of the reasons for the decreasing stress range. The stress range of corn first increased and then decreased with the duration of stress. This was because the developed root system of corn made a high tolerance to the adverse growth environment, resulting in the slow increase of the stress range. But in the later stage, the tall stalks and the large leaf of the corn covered the stress area and generated shadows to affect the temperature extraction by the thermal imager, making a decreasing trend of the stress range on the infrared thermal image.

Combined with the stress range of vegetation in different periods and their mean stress radii, the mean stress radius of grass was the largest. And the stress range showed a trend of steady expansion, which was consistent with the diffusion and accumulation pattern of natural gas leaked into the soil. The large density and the uniform canopy structure of grass and the more obvious stress symptoms make it easier to detect the stress caused by natural gas micro-leakage in the wild. Furthermore, the rhizomes of grass are highly spreading and well adapted to the field environment, which can reduce the time and cost of manual maintenance. Therefore, grass can be used as a good indicator to detect natural gas micro-leakage.

## 4. Conclusions

In this study, infrared thermal images and natural gas concentration data from 2017, 2018, and 2019 were used to evaluate vegetation growth during the natural gas micro-leakage in the Daxing District of Beijing. It was found that the spatial distribution of natural gas in the soil was basically consistent with the temperature features of the stressed vegetation in the infrared thermal images. A deep learning method was proposed to identify stressed vegetation, thereby the area of natural gas micro-leakage can be detected indirectly. The proposed GLNB model for stress detection on grass, corn, soybean and wheat presented the better performance comparing with four classical CNN models. Furthermore, transfer learning was used to pretrain the GLNB model and the result showed that this method improved classification efficiency and detection accuracy. Finally, the range of stressed vegetation under natural gas micro-leakage was located based on GLNB and Grad-CAM. The time series analysis results showed that the generality and high efficiency of the proposed GLNB model made it applicable to multiple periods and various field plants, which overcame the challenge in terms of irregularity in the outdoor environment. However, there is still a scope for improvement. The response of stressed vegetation in canopy temperature is slow, so hyperspectral remote sensing technology can be added to study the spectral reflectance of vegetation for the early detection of natural gas micro-leakage areas. In addition, large-scale detection of natural gas micro-leakage can be realized by UAV technology as a future scope.

## Figures and Tables

**Figure 1 sensors-22-05322-f001:**
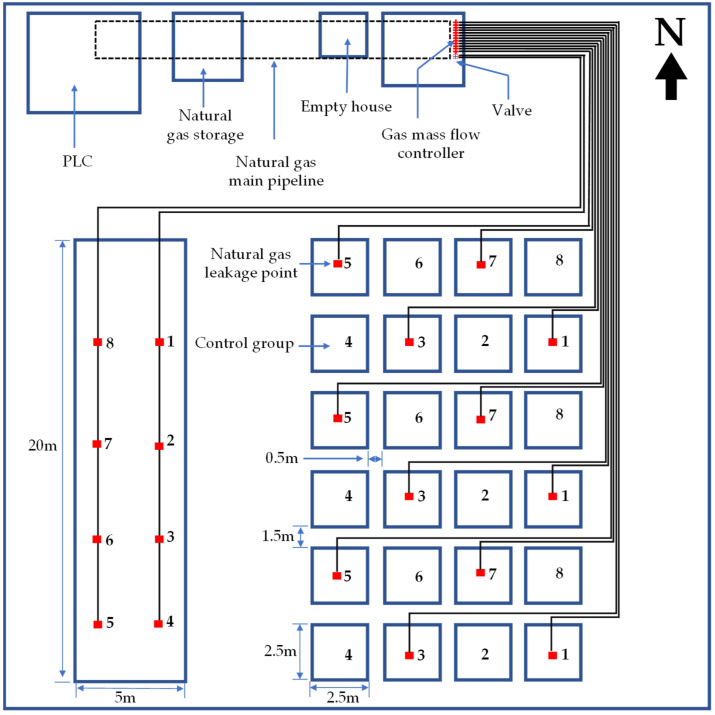
The sketch map of the experimental area; the numbers in a 20 m × 5 m plot are marked leak point, and the 2.5 m × 2.5 m plots with odd numbers represent the experimental plots whereas these with even numbers represent control plots without natural gas.

**Figure 2 sensors-22-05322-f002:**
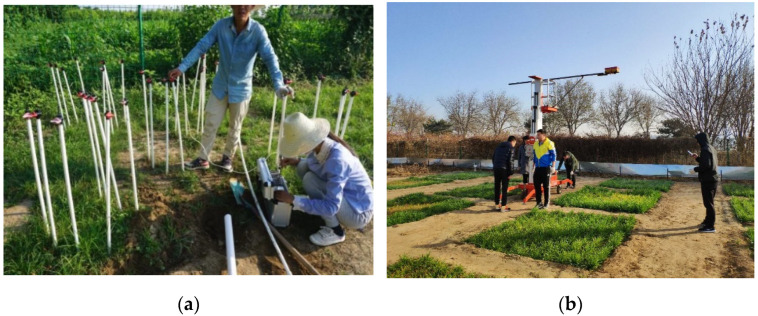
(**a**) Gas concentration data collection using GXH-3050E gas analyzer; (**b**) Infrared thermal images acquisition using Fluke ti55ft thermal imager.

**Figure 3 sensors-22-05322-f003:**
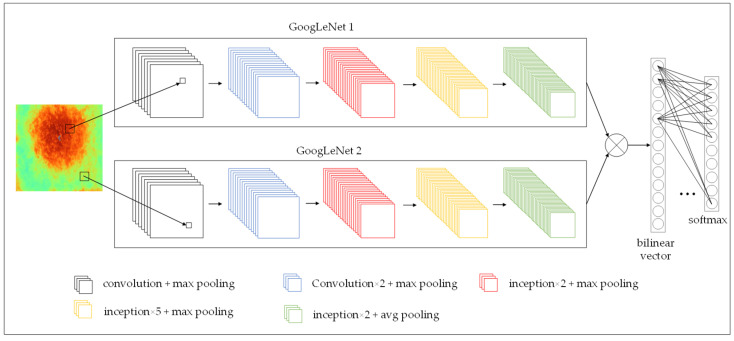
The architecture of the GLNB model.

**Figure 4 sensors-22-05322-f004:**
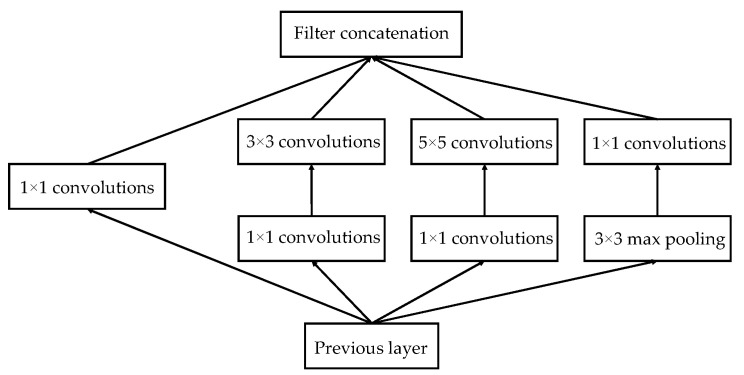
Architecture of the inception.

**Figure 5 sensors-22-05322-f005:**
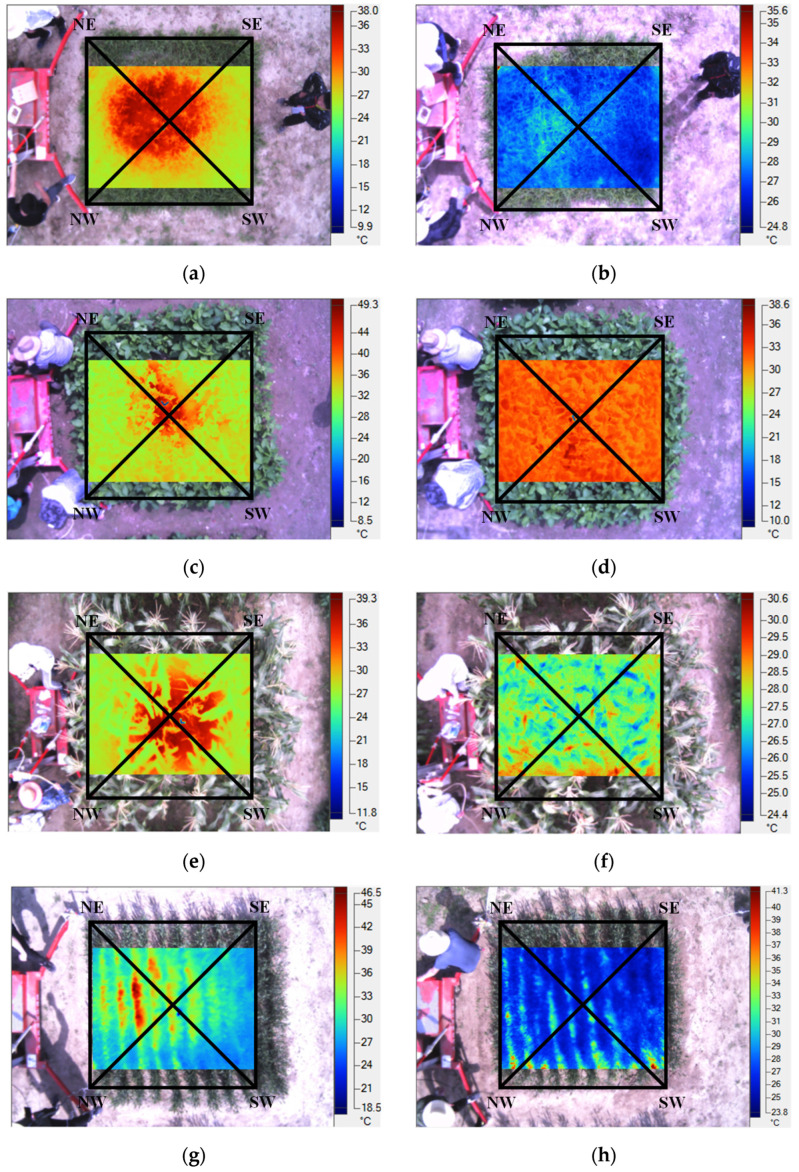
Infrared thermal images of experimental groups: (**a**) grass; (**c**) soybean; (**e**) corn and (**g**) wheat. Infrared thermal images of control groups: (**b**) grass; (**d**) soybean; (**f**) corn and (**h**) wheat.

**Figure 6 sensors-22-05322-f006:**
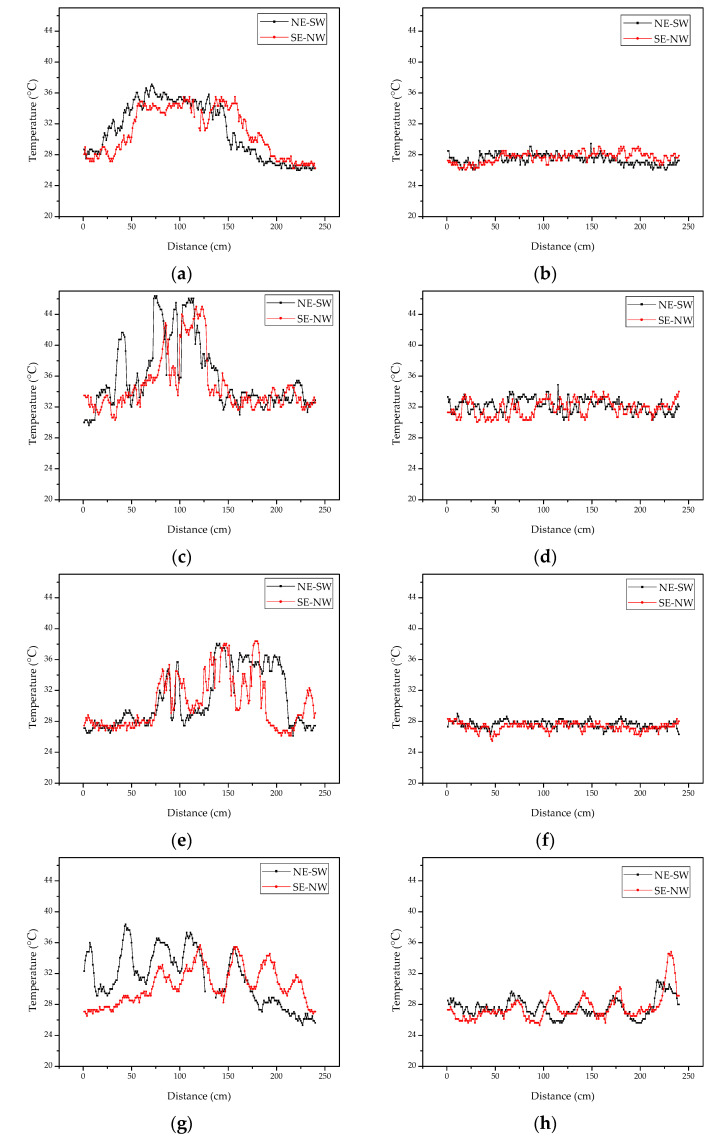
Temperature profiles of experimental groups: (**a**) grass; (**c**) soybean; (**e**) corn and (**g**) wheat. Temperature profiles of control groups: (**b**) grass; (**d**) soybean; (**f**) corn and (**h**) wheat.

**Figure 7 sensors-22-05322-f007:**
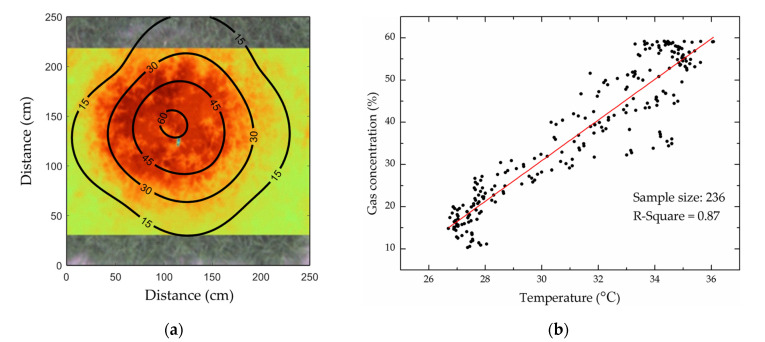
(**a**) Spatial relationship between methane concentration in soil and stressed vegetation areas. (**b**) Correlation analysis results of methane concentration and canopy temperature.

**Figure 8 sensors-22-05322-f008:**
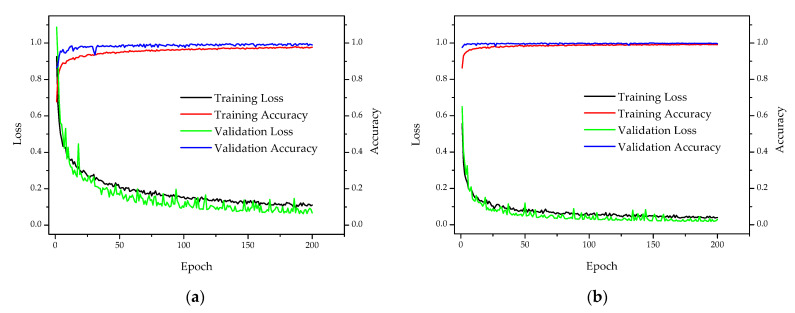
The classification accuracy and cross-entropy loss of the two models: (**a**) the GLNB; (**b**) the GLNB-based transfer learning model (GLNB-TL).

**Figure 9 sensors-22-05322-f009:**
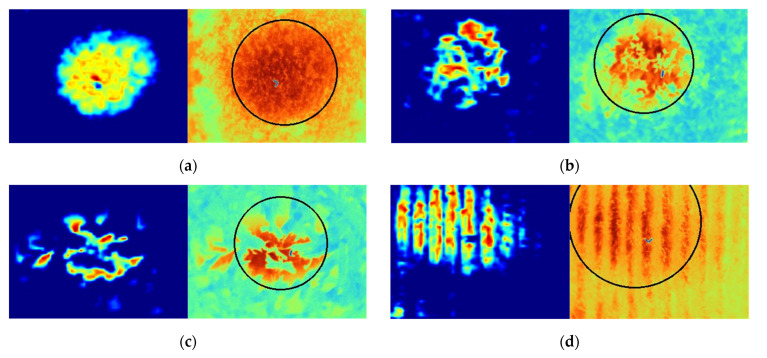
Visualization of the used areas in random input images for classifier: (**a**) grass; (**b**) soybean; (**c**) corn and (**d**) wheat. Each image from left to right is the visual heat map based on Grad-CAM and the heat map fitting circle.

**Figure 10 sensors-22-05322-f010:**
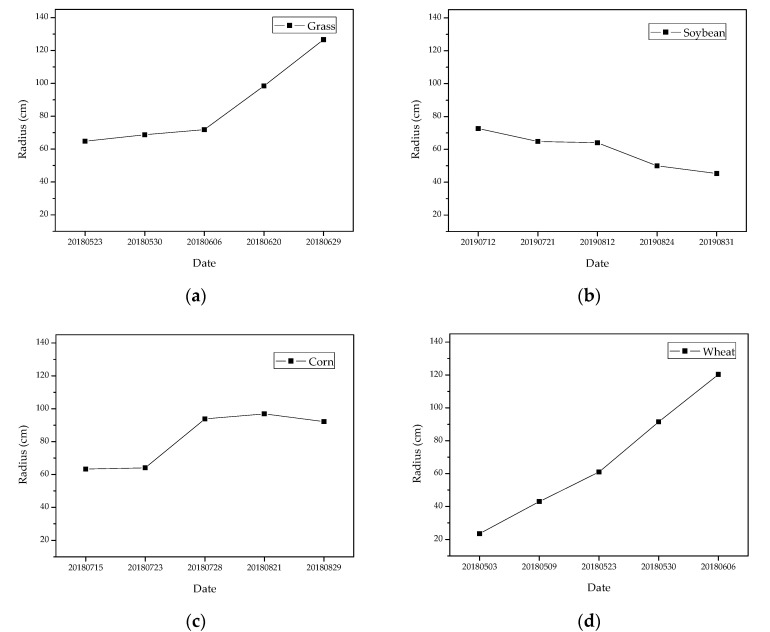
Temporal and spatial variation of stressed vegetation range under natural gas micro-leakage: (**a**) grass; (**b**) soybean; (**c**) corn and (**d**) wheat.

**Table 1 sensors-22-05322-t001:** Statistical characteristic values of temperature in the section of experimental and control plots.

Vegetation Type	Plots	Mean/°C	Standard Deviation/°C	Minimum/°C	Maximum/°C
Grass	Experimental plot	31.07	3.35	26.00	37.13
	Control plot	27.53	0.67	26.06	29.44
Soybean	Experimental plot	35.36	4.07	29.63	46.38
	Control plot	32.02	0.96	30.06	34.88
Corn	Experimental plot	30.56	3.51	26.13	38.38
	Control plot	27.46	0.52	25.44	29.00
Wheat	Experimental plot	30.98	2.97	25.31	38.38
	Control plot	27.58	1.48	25.31	34.81

**Table 2 sensors-22-05322-t002:** Classification efficiency comparison among different CNN models.

Vegetation Type	Evaluation Indicators	AlexNet	VGG16	GoogLeNet	ResNet	GLNB
Grass	Accuracy (%)	94.39	90.65	89.72	91.59	95.33
	Recall (%)	96.15	92.31	88.46	90.38	96.15
	Specificity (%)	92.73	89.09	90.91	92.73	94.55
Soybean	Accuracy (%)	85.05	82.24	86.92	79.44	90.65
	Recall (%)	85.71	76.79	83.93	78.57	89.29
	Specificity (%)	84.31	88.24	90.20	80.39	92.16
Corn	Accuracy (%)	85.98	86.92	81.31	83.18	88.79
	Recall (%)	90.63	87.50	81.25	90.63	90.63
	Specificity (%)	84.00	86.67	81.33	80.00	88.00
Wheat	Accuracy (%)	85.98	90.65	87.85	84.11	91.59
	Recall (%)	88.14	91.53	89.83	86.44	93.22
	Specificity (%)	83.33	89.58	85.42	81.25	89.58
Average per-class	Accuracy (%)	87.85	87.62	86.45	84.58	91.59
	Recall (%)	90.16	87.03	85.87	86.51	92.32
	Specificity (%)	86.09	88.40	86.97	83.59	91.07

**Table 3 sensors-22-05322-t003:** Comparison of the classification performance between GLNB and GLNB-TL.

Vegetation Type	Evaluation Indicators	GLNB	GLNB-TL
Grass	Accuracy (%)	95.33	98.13
	Recall (%)	96.15	98.08
	Specificity (%)	94.55	98.18
Soybean	Accuracy (%)	90.65	96.26
	Recall (%)	89.29	96.43
	Specificity (%)	92.16	96.08
Corn	Accuracy (%)	88.79	91.59
	Recall (%)	90.63	90.63
	Specificity (%)	88.00	92.00
Wheat	Accuracy (%)	91.59	95.33
	Recall (%)	93.22	94.92
	Specificity (%)	89.58	95.83
Average per-class	Accuracy (%)	91.59	95.33
	Recall (%)	92.32	95.02
	Specificity (%)	91.07	95.52

## Data Availability

The data presented in this study are available on request from the corresponding author.

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
