# Peer review of "Deep Learning Approach for Detection of Underground Natural Gas Micro-Leakage Using Infrared Thermal Images"

_sensors, 2022, doi:10.3390/s22145322_

Round 1

Reviewer 1 Report

This draft has described a study about using infrared thermal imaging combined with deep learning to detect natural gas micro-leakage areas. The results are based on the different canopy temperature characteristics of four vegetation varieties under natural gas stress from 2017 to 2019. It is showed that the canopy temperature of vegetation increased under gas stress. A GoogLeNet model with Bilinear pooling (GLNB) was proposed for the classification under natural gas micro-leakage stress and transfer learning was used to improve the model training process and classification efficiency. Authors achieved 95.33% average accuracy, 95.02% average recall and 95.52% average specificity of stress classification for four vegetation varieties and determined the range of natural gas micro-leakage stress areas under different vegetation and stress durations. The draft has shown a promising study using deep learning to detect gas leakage and it should be interesting for the readership of Sensors. However, I  have some concerns before I suggest this paper to be accepted.

1.       On line 117, the total pixel numbers are not resolution. The authors should clarify this. Also, what is the temperature resolution of thermal image?

2.     There are some wrong with Fig. 2, 7 ,8 and 9. The captions are totally messed up.

3.     Fig. 5 is not organized well and Fig.6 is too difficult to read.

4.     The images are taken in cloudless condition, but how about the daily temperature? Will the daily temperature variation affect the results?

5.     For the temperature difference between healthy vegetation and stressed vegetation, it can result from many reasons as mentioned in the introduction. But will they have similar affects to that due to gas leakage? Can these factors be separated by the method of the paper?

Reviewer 2 Report

In general, the presented work is well-written and well organized and worth publishing.

However, it still has some issues that need to be addressed.

1. As the research work proposed to use some transfer learning method to improve the accuracy, I suggest the authors could add some literature analysis about the transfer learning method for detection, such as:

https://www.sciencedirect.com/science/article/pii/S0888327021000133

https://www.sciencedirect.com/science/article/pii/S089360802030229X

https://ieeexplore.ieee.org/abstract/document/8713860

2. In the results and analysis, the selection hyper-parameters should be discussed in detail.

3. The resolution of Figure 6 should be increased, and the size of the axis number should be enlarged.
